# The LOLland offshore Lidar EXperiment (LOLLEX): A novel observational approach for the study of wind farm flow and entrainment

Shokoufeh Malekmohammadi<sup>1,2</sup>, Etienne Cheynet<sup>1,2</sup>, Joachim Reuder<sup>1,2,3</sup>, Claus Linnemann<sup>4</sup>, Mikael Sjöholm<sup>5</sup>, Jakob Mann<sup>5</sup>, and Gregor Giebel<sup>5</sup>

Correspondence: Shokoufeh Malekmohammadi (shokoufeh.malekmohammadi@uib.no)

#### Abstract.

Vertical momentum entrainment above offshore wind farms plays a key role in the recovery of wind turbine and wind farm wakes but remains poorly documented by field measurements. The LOLland offshore Lidar EXperiment (LOLLEX) campaign introduced a novel measurement approach to address this knowledge gap. The primary objective of this campaign was to develop a new atmospheric measurement strategy to characterise and quantify the vertical momentum entrainment inside and outside an offshore wind farm using Doppler wind lidar technology. LOLLEX was conducted from September 2022 to September 2023 in Denmark in and around the Rødsand II wind farm just south of the island of Lolland. During this campaign, two pulsed Doppler wind lidars, a scanning and a profiling instrument, were deployed onboard a crew transfer vessel (CTV) commuting daily between the harbour and the offshore wind farm Rødsand II. Additionally, a scanning pulsed Doppler wind lidar was mounted on a transformer platform north of the wind farm to perform range height indicator scans across the farm. Motion-corrected mean wind speed data were collected up to 300 m above the sea surface by the profiler lidar. The scanning lidar collected data up to 2.5 km alternating between the profiling mode and vertical stare mode. The latter scan operated with a sampling frequency of 1 Hz and along-beam spatial resolution of 10 m, allowing for the study of the turbulent vertical wind velocity component. The dataset includes several thousand hours of vertical scans. As a result of the moving vessel, many of the observations occurred inside or in the close vicinity of the wind farm, providing insight into the near and far wakes of individual and multiple turbines. The potential and limitations of the new measurement strategy is illustrated using four case studies:(1) the observation of a Kelvin-Helmholtz instability above the wind farm, examined further in a companion paper; (2) turbulent mixing propagating downward from the top of the boundary layer, enhancing momentum entrainment; (3) internal atmospheric waves and (4) wake characterisation inside the wind farm using the range-height indicator scans from the lidar deployed on the platform. This work demonstrates a novel methodology integrating remote sensing with a mobile offshore platform to measure turbulence at unprecedented altitudes. The dataset offers valuable data for wind energy research, boundary-layer meteorology, and further development of atmospheric measurement techniques.

<sup>&</sup>lt;sup>1</sup>Geophysical Institute, University of Bergen, Allegaten 70, 5007 Bergen, Norway

<sup>&</sup>lt;sup>2</sup>Bergen Offshore Wind Centre, University of Bergen, Allegaten 55, 5007 Bergen, Norway

<sup>&</sup>lt;sup>3</sup>Bjerknes Centre for Climate Research, Jahnebakken 5, 5007 Bergen, Norway

<sup>&</sup>lt;sup>4</sup>RWE Offshore Wind GmbH, Essen, Germany

<sup>&</sup>lt;sup>5</sup>Department of Wind and Energy Systems, Technical University of Denmark, Frederiksborgvej 399, 4000 Roskilde, Denmark.

#### 1 Introduction

25

The European Union aims to commission at least 300 GW of offshore wind capacity by 2050, compared to 19 GW in 2023 (European Commission, 2023). Achieving this target requires a significant increase in the speed and scale of turbine deployment. Due to limited available space in the North and Baltic Seas, wind farms are being built in increasingly dense configurations, intensifying turbine interactions and wake effects. Wake losses refer to the reduction in power output caused by turbine interactions within a wind farm or between farms. Within a farm, downstream turbines operate in the slower, more turbulent air generated by upstream turbines. These wake effects reduce annual energy production (AEP), complicate farm layout design, and thus increase the levelized cost of energy (LCOE) (Akhtar et al., 2021; Pryor et al., 2021; Minz et al., 2025). They may also influence legal frameworks (Finserås et al., 2024) and affect electricity market integration (Kenis et al., 2023). Coupled mesoscale-microscale wind models and analytical wake models (AWMs) have been widely used to quantify wake losses, assess wake extent, and support AEP and LCOE optimisation (Sanz Rodrigo et al., 2017; Pryor et al., 2021; Haupt et al., 2023; Wang et al., 2024).

Beyond a certain scale, wake recovery is governed primarily by vertical momentum entrainment from the atmospheric boundary layer (ABL) into the internal boundary layer (IBL), which replenishes energy within the farm (Frandsen, 1992; Abkar and Porté-Agel, 2013; Calaf et al., 2010; Porté-Agel et al., 2020; Meneveau, 2019; Krishnamurthy et al., 2025; Bempedelis et al., 2023), as sketched in Fig. 1. However, current wind farms seem not to have reached this scale and cannot be considered infinite (Syed et al., 2023). Vertical momentum entrainment has been modelled for both finite and idealised (infinite) wind farms, with validation from Large Eddy Simulations (LES) and limited field data (Calaf et al., 2010; Luzzatto-Fegiz and Caulfield, 2018; Bempedelis et al., 2023). However, direct observations of this process in offshore environments remain scarce (Meneveau, 2019; Giebel et al., 2021; Syed et al., 2023; Krishnamurthy et al., 2025).

Accurately capturing entrainment and wake recovery processes requires advanced observational tools. Remote sensing has played a key role, with Doppler wind lidar (DWL) (Käsler et al., 2010; Krishnamurthy et al., 2017), Doppler radar (Hirth et al., 2015; Ahsbahs et al., 2020), airborne measurements (Syed et al., 2023), and synthetic aperture radar (Platis et al., 2018; Ahsbahs et al., 2020; Finserås et al., 2024) all providing valuable data.

In wind energy science, commercially available DWL profilers and scanners are commonly used. Profiler lidars are primarily deployed to measure mean wind speed profiles in the first 300 m above the surface. In contrast, long—range scanning lidars can provide additional insights into mean and turbulent flow characteristics with ranges extending several kilometres. The scanning DWLs can operate in various modes, including Doppler beam swinging (DBS) for wind profiling, range-height indicator scans, plan-position indicator, and fixed line-of-sight scans. Yet most offshore DWL deployments rely on fixed platforms like met masts or substations (Krishnamurthy et al., 2017), limiting spatial coverage. Although profiler lidars have been deployed on floating buoys or vessels (Gottschall et al., 2017, 2018; Kelberlau and Mann, 2022; Rubio et al., 2022), they are generally used to

Figure 1. Schematic of vertical momentum entrainment from the ABL into the IBL of a wind farm.

measure the mean wind speed at limited heights. Scanning lidar deployments on vessels or buoys are rare, despite their potential to capture the structure of the atmosphere above offshore wind farms, including vertical momentum entrainment (Abkar and Porté-Agel, 2013; Luzzatto-Fegiz and Caulfield, 2018; Syed et al., 2023).

To address this gap, we present a new measurement strategy using two DWLs mounted on a crew transfer vessel (CTV). This dual-lidar approach combines a lidar wind profiler and a scanning wind lidar in vertical stare mode. This mobile setup allows flexible positioning near or inside the wind farm, providing wind speed profiles and high-resolution vertical velocity measurements. This combination provides enhanced capability for studying the wake recovery and vertical momentum entrainment compared to a fixed lidar or a ship-based lidar wind profiler.

This paper provides an overview of the LOLland offshore Lidar EXperiment (LOLLEX), conducted from September 2022 to September 2023 at the Rødsand II wind farm in Denmark and presents the potential of the collected dataset in the form of selected case studies. It starts with the campaign description in Section 2, including site characterisation, instrumentation and the selected lidar measurement setup. Section 3 describes the data processing workflow, including motion compensation, data filtering, and the complementary use of the mesoscale wind database NORA3 alongside remote sensing observations. Furthermore, Section 4 summarises the collected data and its availability, and Section 5 presents four case studies, three based on the scanning lidar mounted on the CTV and one from the scanning lidar on the transformer platform. Finally, Section 6 evaluates the potential and limitations of the current measurement setup in comparison with DWLs deployed on fixed platforms or buoys.

# 2 Campaign description

The LOLLEX measurement campaign was carried out between September 2022 and September 2023 as a collaborative effort between the MSCA-ITN project Train<sup>2</sup>Wind, funded by the EU Horizon 2020 scheme, and the RWE group. The main goal of Train<sup>2</sup>Wind was to advance the understanding of entrainment processes relevant for large offshore wind farms. The choice of Rødsand II was mainly guided by logistical considerations, as the accessibility of a well-suited CTV and the well-positioned transformer platform. Its proximity to the land was an additional factor, as the initial experiment plan also included considerable complementary measurement activities using various uncrewed aerial systems. However, most of these activities could not be performed as intended due to regulatory challenges.

#### 2.1 Experiment site





The Rødsand II wind farm (54.5799°N, 11.8903°E), in operation since 2010, is located in the shallow Danish waters of the Baltic Sea, south of the island of Lolland and close to the German border (Fig. 2). The farm consists of 90 Siemens SWT–2.3–93 turbines, each with a nominal capacity of 2.3 MW, a hub height of 68.5 m, and a rotor diameter (D) of 93 m. The farm is arranged in five curved rows of 18 turbines each, with inter–turbine spacing ranging between 5D and 8D (see also the bottom panel of Fig. 5). Approximately 3 km to the east lies the Nysted wind farm, commissioned in 2003, comprising 72 Bonus 2.3 MW turbines with a spacing ranging from 6D to 10D. The nearest landmass, the island of Lolland, is characterised by low-lying, flat terrain with sandy beaches, coastal dunes, agricultural fields, small settlements, and patches of forest. The region experiences high average wind speeds of about 9.6 m s<sup>-1</sup> at 100 m, predominantly from the west to the south-west, influenced by weather systems passing over the North Sea and Baltic Sea (Davis et al., 2023, see also https://globalwindatlas.info/en/).

During more than a decade, Rødsand II and Nysted have been used for studies on the interaction between atmospheric flow and wind farm arrays, in particular wind turbine wakes and wind farm interactions, also called cluster effect (Hansen et al., 2015). Previous studies focusing on wake loss for these two farms used simulations alone (Nygaard and Hansen, 2016), combined SCADA data and numerical simulations (Hansen et al., 2015; Fischereit et al., 2022) or simulation and mast-based measurements (Cleve et al., 2009).

There have, however, been only a few measurement campaigns that have focused on wake flow characteristics and the structure of the ABL above Rødsand II. In 2013–2014, a scanning lidar campaign under the lead of the Carbon Trust Offshore Wind Accelerator program was performed. It was aimed at investigating wake flow and boundary layer dynamics at Rødsand II. Only a limited portion of the results from this campaign has, however, been openly published (Adams, 2015).

The LOLLEX campaign builds on these foundations and introduces several novel and complementary elements: a long-range scanning lidar deployed on the offshore transformer platform north of the wind farm and a dual scanning-profiling lidar setup mounted on a CTV as an observational platform of opportunity. The WindCube lidars were provided by the national Norwegian research infrastructure OBLO (Offshore Boundary Layer Observatory) operated by the University of Bergen (UiB), and the lidar on the transformer platform was supplied by the Technical University of Denmark (DTU Wind and Energy Systems).

**Figure 2.** The location of the Rødsand II wind farm in the shallow waters of the Baltic Sea, south of Denmark, near the island of Lolland. This wind farm is located adjacent to the Nysted wind farm. This map was created using Rasterio (an open-source Python library by MapBox) version 1.4.2 (https://rasterio.readthedocs.io/). The digital elevation model was obtained from SRTM data V4, provided by the International Centre for Tropical Agriculture (CIAT), available at https://srtm.csi.cgiar.org (Jarvis et al., 2008).

To complement the lidar data, high-resolution mesoscale wind speed data from the 3-km Norwegian Hindcast Archive (NORA3) (Haakenstad et al., 2021) were used. NORA3 is a state-of-the-art hindcast dataset produced by dynamically downscaling ERA5 reanalysis data (Hersbach et al., 2020). It provides hourly wind conditions over Northern Europe at a

horizontal resolution of 3 km. The dataset has been validated against both in-situ observations (Solbrekke et al., 2021) and remote sensing data (Cheynet et al., 2025), demonstrating excellent performance in coastal and offshore environments.

During the LOLLEX campaign, a specific subset of the NORA3 database, containing wind speed data at seven vertical levels from 10 m to 750 m, was used in combination with the lidar observations above the wind farm, as shown in Malekmohammadi et al. (2025). It was also used to complement the Range Height Indicator (RHI) scans from the lidar installed on the transformer platform (see section 5).

#### 2.2 Instrumentation





The backbone of the campaign was the installation of two DWLs onboard a CTV serving the Rødsand II wind farm, using it as an observational platform of opportunity. This mobile measurement set—up over several months was complemented by the shorter fixed deployment of an additional scanning lidar system on the transformer platform of the wind farm.

#### 2.2.1 Lidar deployment on the CTV

In September 2022, the WindCubeV2 was installed on the stern of a 27 m long and 10 m wide CTV, which operated nearly daily between Rødby harbour and the Rødsand II wind farm for the transport of personnel and material for maintenance purposes. The vessel operations are typically performed between 7:00 am and 7:00 pm, providing up to 12 hours of wind field observations offshore, inside and around the farm. The lidar was installed with its east–facing side oriented toward the bow. From September 2022 to January 2023 the WindCube100S remained onshore, deployed in the parking area of RWE Wind Services Denmark at the quay in Rødbyhavn, allowing for night–time co–located measurements while the CTV was in harbour. The second phase of the LOLLEX campaign started in the last week of January 2023 with the installation of the WindCube100S on the CTV side-by-side with the WindCubeV2 (Fig. 3).

The WindCubeV2 measured wind speed and direction at 11 heights, ranging from 40 m to 290 m, with a sampling frequency of 0.25 Hz. The system included a motion sensor and an Inertial Measurement Unit (IMU) that recorded attitude data at 10 Hz. The IMU tracked pitch, roll, yaw, and translational velocities in the x, y, and z directions, as well as GPS coordinates (latitude, longitude, altitude). It also provided the vessel's heading using GPS True Heading.

The WindCube100S is a scanning lidar with hemispherical scanning capability. It operated at 1 Hz sampling frequency, at ranges between 50 m and about 2500 m. The instrument followed a 30-minute scan cycle alternating between two modes. At full and half hour, it performed 5 minutes of Doppler Beam Swinging (DBS) for wind profiling (right panel in Fig. 4), in between it collected for 25 minutes the vertical velocity component in vertical stare mode (left panel in Fig. 4). This schedule provided an appropriate balance between robust turbulence statistics and reliable mean wind profiles with sufficient temporal resolution. In DBS mode, the WindCube100S retrieved wind profiles up to 2 km above the surface, clearly exceeding the range of the continuous profiling WindCubeV2. In vertical stare mode, it measured vertical wind velocity by emitting a beam at a 90° elevation angle. The range gate length was set to 25 m with 60% geometric overlap, resulting in an effective range resolution of 10 m.

**Figure 3.** Location of the WindCubeV2 and WindCube100S lidars mounted on the stern of the CTV. Azimuth angles were defined relative to the vessel's heading and corrected using GPS true heading from the IMU.

Figure 4. The scanning pattern of WindCube 100S in vertical stare mode (left) and DBS mode (right).

Both instruments were remotely accessible, and the lidar data were uploaded to a server daily. Field inspections and basic maintenance took place regularly, and during inspections, wind and motion data were also collected directly from the lidar software. As the scanning lidar does not support near real-time motion-correction, motion-correction algorithms were applied during post-processing using the IMU data for both systems (see Section 3.3).

**Figure 5.** Top panel: The scanning lidar Halo Photonics by Lumibird StreamLine XR+ installed on the substation (view toward South). Bottom panel: Sketch of the lidar line of sight (top view) with an azimuth angle of 206° and a scanning range of 6 km.

#### 2.2.2 Scanning lidar on transformer platform

To complement the ship—based measurements, a Halo Photonics by Lumibird StreamLine XR+ scanning Doppler lidar was deployed on the transformer platform located north of the wind farm from April to July 2023 (Fig. 5). The lidar was mounted at a height of approximately 25 m above sea level. It operated in RHI mode with a fixed azimuth angle of 206° and elevation angles increasing from –1°to +10°, at a 1°/s rate with an accumulation time of typically 0.5 s. The range gate length was 12 m, and a full RHI scan, including a fast return, took about 16 seconds.

The scan sector extended from north to south across the wind farm. This allowed for the observation of wake propagation during both westerly and easterly wind events. When the wind direction was approximately perpendicular to the lidar's line of sight, the nearest turbine was about 0.7 km away. At that distance, the lidar beam intersected the rotor area around 20 times per scan, yielding a vertical resolution of approximately 6 m within the wake. The farthest turbine was located 3.8 km away, where the beam crossed the rotor area only three times per scan direction, resulting in a vertical resolution of about 38 m (Fig. 6).

At longer distances, this decrease in vertical resolution reduced the accuracy of wake characterisation. The exact wake position and shape also varied with wind direction and turbine operating conditions, such as yaw misalignment and rotor tilt.

#### 3 Methods


This section describes the motion correction algorithm applied to the data collected by the lidar wind profiler WindCubeV2 deployed on the CTV. It also introduces a new method to correct for the static tilt angle error of the WindCube100S in vertical stare mode. Finally, it details the pre-processing steps for filtering the data and dismissing the erroneous data

**Figure 6.** Sketch of an RHI scan in the case where the wake direction is perpendicular to the azimuth. The turbine rotors are indicated as grey circles. The inset sketches the lidar beams crossing the rotor plane at different distances: at around 3.8 km (top inset) and 0.7 km (bottom inset) from the lidar.

# 160 3.1 Data processing and quality control


In a first processing step, all lidar measurements were subject to standard quality control procedures, as signal-to-noise filtering and outlier removal, to identify high-quality data for further analysis. For the ship-based lidar systems, additional motion correction was performed.

Both, the WindCubeV2 and WindCube100S lidars provide a Carrier–to–Noise Ratio (CNR), which is commonly used to assess signal quality. Measurements were filtered using a CNR threshold applied at each location and time step. Following earlier studies (Kumer et al., 2014; Cheynet et al., 2017b; Suomi et al., 2017), a threshold of -24 dB was used for the WindCubeV2, and -27 dB for the WindCube100S. When deployed on the vessel, the WindCube100S occasionally produced short or incomplete scans, likely due to its sensitivity to rapid translation and acceleration. Unlike the WindCubeV2, which is designed for dynamic conditions on buoys, the WindCube100S is not optimised for mobile offshore deployments. Despite these limitations, a large number of high-quality scans were obtained, providing valuable measurements within the wind farm. Outlier removal was accomplished by a moving median absolute deviation filter, following the recommendation by Starkenburg et al. (2016) and Leys et al. (2013). Following filtering and outlier removal, motion correction was applied to the WindCubeV2 data and to the DBS–mode scans of the WindCube100S, as described in Section 3.2.


The radial velocity data from the Halo Photonics StreamLine XR+ operating in RHI mode is filtered using the signal-to-noise ratio (SNR) percentiles and the velocity distribution. First, values with SNR above the 99th percentile or below the 1st percentile are removed. Then, a probability density function of the remaining velocity data is estimated. The most frequent (peak) velocity is identified from the probability density function. Only velocity values within a band around this peak are kept. The bandwidth is set to  $\pm 8$  m s<sup>-1</sup> around the peak velocity. All data outside this range are set to NaN and excluded from further analysis.

## 3.2 Motion correction for lidar wind profiler

A lidar wind profiler mounted on a moving platform, such as a buoy or ship, is subject to six degrees of freedom (DOF) of motion. These include translational movements, such as surge, sway, and heave along the platform's x, y, and z axes, as well as rotational movements, such as roll  $(\beta)$ , pitch  $(\varphi)$ , and yaw  $(\psi)$  around these respective axes. These motions alter the measurement geometry and introduce artificial velocity components that affect both the observed radial wind speed  $(v_r)$  and the reconstructed three-dimensional wind vector  $(\hat{u})$ .

For a stationary platform, the 10-minute average wind speeds are only marginally impacted (typically by 1–2%) due to averaging over wave-induced quasi-random motion (Malekmohammadi et al., 2024; Kelberlau et al., 2020). However, instantaneous measurements can be significantly affected. Therefore, robust motion correction is essential when high–resolution wind data is required, especially in offshore environments or for applications such as wake studies or entrainment characterisation.

Several motion correction algorithms have been developed to compensate for the motion-induced distortions in lidar measurements from floating platforms (Peña et al., 2013; Kelberlau et al., 2020; Wolken-Möhlmann et al., 2014; Bischoff et al., 2015). These efforts primarily target periodic wave-induced motions, typically observed in buoy-based systems. However, ship-based systems experience both rotational and non-periodic translational motion patterns that can be challenging to correct with standard correction techniques.

In this study, we adopt the correction method described by Malekmohammadi et al. (2024), which is suitable for periodic and aperiodic motion conditions. The algorithm corrects each lidar beam's line-of-sight (LOS) velocity in two steps: (1) Translational Motion Correction: The platform's instantaneous translational velocity vector  $u_T$ , comprising surge, sway, and heave, is projected onto the beam direction and subtracted from the measured  $v_r$ ; (2) Rotational Motion Correction: At each time step, the platform's orientation is used to construct a composite rotation matrix  $\mathbf{R}$  based on the roll, pitch, and yaw angles. This matrix transforms the wind vector into the lidar's moving coordinate frame as

$$\boldsymbol{v_r} = \mathbf{RN}(\boldsymbol{u} + \boldsymbol{u_T}), \tag{1}$$

where  $v_r$  is the vector of observed radial velocities, u is the true wind vector in the inertial frame,  $u_T$  is the translational motion vector, and  $\mathbf{R}\mathbf{N}$  is the product of the rotation matrix  $\mathbf{R}$  and the beam direction matrix  $\mathbf{N}$ . The matrix  $\mathbf{R}$  is computed

from the time-resolved attitude data using standard Euler rotations:

$$R_x = \begin{pmatrix} 1 & 0 & 0 \\ 0 & \cos \beta & \sin \beta \\ 0 & -\sin \beta & \cos \beta \end{pmatrix},\tag{2}$$

$$R_{x} = \begin{pmatrix} 1 & 0 & 0 \\ 0 & \cos \beta & \sin \beta \\ 0 & -\sin \beta & \cos \beta \end{pmatrix},$$

$$R_{y} = \begin{pmatrix} \cos \varphi & 0 & -\sin \varphi \\ 0 & 1 & 0 \\ \sin \varphi & 0 & \cos \varphi \end{pmatrix},$$

$$(2)$$

$$R_z = \begin{pmatrix} \cos\psi & \sin\psi & 0 \\ -\sin\psi & \cos\psi & 0 \\ 0 & 0 & 1 \end{pmatrix}. \tag{4}$$

These are combined as  $\mathbf{R} = R_z R_y R_x$  to account for the platform's full attitude. The corrected wind vector is then obtained by least-squares inversion:

$$\hat{u} = [(\mathbf{R}\mathbf{N})^T \mathbf{R}\mathbf{N}]^{-1} (\mathbf{R}\mathbf{N})^T (\mathbf{v_r} - \mathbf{v_{rT}})$$
(5)

where  $v_{rT} = \mathbf{RN}u_T$ .


Unlike earlier approaches limited to sinusoidal or low-frequency oscillations, the method applied here accommodates the erratic and multi-directional motions typical of ship-based lidar deployments. The efficacy of this algorithm has been validated under controlled conditions using a six-DOF motion platform (Malekmohammadi et al., 2024). Thus, this motion correction approach is well-suited for the deployment of DWLs on CTV or other vessels.

#### 215 3.3 Static tilt correction for scanning lidar

In this study, lidars operating on a moving platform in DBS mode emit four to five beams, which result in four to five equations for the three unknown wind components (u, v, and w). Due to this redundancy, the retrieved wind speeds can be corrected for motion. Nevertheless, a correction of this type is not possible in the vertical stare mode since there is only one equation with three unknowns in the system of equations.

In Malekmohammadi et al. (2025), a method to correct for static tilt angle error was outlined and applied to a single 30-minute record. In the present study, we extend this method to multiple records to assess its robustness. We compare it to a simpler tilt angle correction method based on the assumption of a mean horizontal flow at 400 m above the surface, as well as to the case where no correction is applied. For completeness, the method is summarised in the following and its performance is discussed.

In vertical stare mode, the LOS velocity component closely approximates the vertical velocity component, unless non-zero tilt angles are present. The static (time-averaged) tilt angle can be estimated and used to correct the lidar-retrieved vertical velocity 225 component. The method is inspired by the double rotation algorithm commonly used for tilt correction of sonic anemometers. In the present case, the equation is given by

$$\overline{w}_c = -\overline{u} \cdot \operatorname{sign}(\overline{w}_{\operatorname{uncor}}) \cdot \sin(\theta) + \overline{w}_{\operatorname{uncor}} \cdot \cos(\theta), \tag{6}$$


Figure 7. Five-minute moving average of the vertical (top) and horizontal (bottom) wind velocity components measured by the WindCubeV2 and WindCube100S at 150 m above the surface. The top panel compares three tilt correction approaches for the vertical component: a correction using  $\theta = 2.7^{\circ}$  (Eq. 6), a method assuming zero mean vertical velocity at 400 m, and no correction.

where  $\overline{w}_c$  is the corrected vertical velocity component at height z,  $\overline{u}$  is the mean horizontal wind speed at the same height,  $\overline{w}_{uncor}$  is the uncorrected vertical velocity component recorded by the lidar (i.e., the along-beam velocity), and  $\theta$  is the static tilt angle. The unknown in this equation can be  $\overline{w}_c$ ,  $\overline{u}$ , or  $\theta$ , the latter being independent of height. The angle  $\theta$  can be determined either through visual inspection of the lidar's inclinometer or estimated using  $\overline{w}_c$  and  $\overline{u}$  from the lidar wind profiler WindCubeV2, as done in Malekmohammadi et al. (2025). Both visual inspection and the application of eq. (6) using data from WindCubeV2 yielded a static tilt angle of  $\theta = 2.7^\circ$ .

Figure 7 compares the vertical wind velocity from the WindCube100S using three different methods: no correction, a correction assuming  $\overline{w} = 0$  at 400 m, and tilt correction with  $\theta = 2.7^{\circ}$  (Eq. 6). The case from 21 February 2023 was selected because both the WindCubeV2 and WindCube100S provided high–quality data, with excellent agreement in mean wind speed at 150 m.

The method based on Eq. 6 shows the best match with the mean vertical velocity measured by the WindCubeV2 lidar. The approach assuming zero vertical wind at 400 m performs less well but still significantly reduces the bias. Without any correction, the vertical velocity from the scanning lidar shows a bias of up to 1 m s<sup>-1</sup>. The WindCube100S lidar is used here instead of the WindCubeV2 because it samples five times faster and measures up to 2.5 km, compared to 300 m for the WindCubeV2. Hereinafter, the scanning lidar data are shown with tilt correction using Eq. 6. While the mean velocity component is typically

**Figure 8.** Periods of available data for the three lidars during the LOLLEX campaign: the WindCube100S scanning lidar (light blue when onshore, dark blue when on the CTV), the WindCubeV2 wind profiler lidar on the CTV (green), and the Halo Photonics StreamLine XR+ scanning lidar on the transformer platform (orange).

removed in turbulence analysis, correcting for static tilt bias is still essential here to avoid misinterpreting physically meaningful vertical motions, such as those caused by entrainment or convection.

# 4 Data overview





This section provides an overview of the data collected during the LOLLEX measurement campaign. The campaign lasted about one year and produced approximately 6090 hours of WindCubeV2 measurements and 4500 hours of WindCube100S measurements, as shown in Figure 8. The dataset includes periods when the CTV was located in the harbour, in transit to or from the wind farm, and during mobile and stationary operations inside the wind farm. Approximately nine thousand RHI scans of 14 min were collected by the scanning lidar on the transformer platform, yielding over 2100 hours of data.

#### 4.1 Data availability

Figure 8 shows the data availability of the WindCube100S, WindCubeV2, and the Halo Photonics by Lumibird StreamLine XR+ scanning lidar on the transformer platform throughout the campaign. For the WindCube100S, high data availability was recorded from September 2022 to July 2023, except during the period between December 2022 and January 2023 when the lidar was being installed on the CTV. The WindCubeV2 exhibited even higher availability, with only one significant gap in April 2023. The scanning lidar on the platform recorded reliable data from May to August 2023. As a result, overlapping measurements from all lidar systems were available between May and July 2023 and again at the end of August 2023.

In 2022, both lidars were located in the harbour but had not yet been installed on the vessel. Beginning in January 2023, the WindCubeV2 and WindCube100S were mounted on the CTV, allowing for direct measurements within the wind farm. From January to August 2023, the CTV operated primarily in the harbour during night and inside the wind farm during daytime, as illustrated by the vessel density map in Fig. 9. This figure uses a pseudocolour plot to visualise the number of hours the CTV


**Figure 9.** Spatial density map of the CTV's position during the LOLLEX campaign. The colour scale indicates the frequency of recorded positions, with darker areas representing locations where the CTV spent more time. Red circles mark the Rødsand II wind turbines. The cross marks the approximate location of Rødbyhavn harbour. The map is based on data recorded by the WindCubeV2 's internal IMU.

spent in each location. The map reveals several high-density clusters, some exceeding 50 hours, near individual wind turbines and the transformer platform. These high-density areas correspond to the turbines most frequently visited for operation and maintenance, as well as offshore idle time. Most turbines were visited at least once, ensuring that lidar measurements were collected across the entire wind farm during stationary periods. This widespread spatial coverage is a clear advantage of the vessel-based lidar setup. Such extensive spatial sampling would not have been achievable using wind lidars mounted only on buoys or fixed platforms.

Figure 10 presents the wind rose based on NORA3 data at 150 meters above sea level. To improve clarity, only wind speeds above 5 meters per second are included. The wind rose is based on 5623 hours of data. Both NORA3 and the WindCubeV2 show that westerly winds dominate, with a median wind speed between 8 and 9 m s<sup>-1</sup> at 150 meters altitude. The WindCubeV2 data indicate more frequent north-easterly winds and stronger winds from the sector between 210 and 260 degrees than NORA3 suggests. Differences in wind direction between the NORA3 dataset and the WindCubeV2 measurements can be explained by two factors. First, the CTV carrying the lidar was not fixed in space and often remained near the harbour or inside the wind farm, while NORA3 data corresponds to a fixed offshore grid point. Second, the NORA3 model does not represent the effects of the Rødsand II and Nysted wind farms on the local wind field.

**Figure 10.** Wind rose for the mean wind speed at 150 m obtained with NORA3 (left) and WindCubeV2 (right) during the Lollex measurement campaign, from 2022-10-09 to 2023-07-12 (6624 h of data).

## 4.2 Time series analysis





This section compares the wind speed data from the two lidars mounted on the CTV with the NORA3 reanalysis. The goal is to assess the overall quality of the lidar measurements and evaluate the consistency between the lidar data and NORA3. This comparison is important to build confidence in the case studies presented later in the paper.

Figure 11 shows the time series of wind speed at 100 m above the surface from NORA3, the WindCubeV2 profiler, and the WindCube100S scanning lidar for the years 2022 and 2023. The average wind speed across all datasets at 100 m height is close to 10 m s<sup>-1</sup>. The WindCubeV2 shows greater variability, as it provides wind speed values every 10 minutes, while NORA3 outputs wind speed only once per hour, even though it represents a 10-minute average centred around the full hour. The WindCube100S collected DBS measurements every 30 minutes, based on 5-minute scan periods, leading to irregular time series sampling.

To perform a fair comparison, all data must be expressed using the same averaging time. NORA3 and WindCubeV2 already use 10-minute averages. The WindCube100S uses 5-minute averages, which must be adjusted. To account for this difference, we apply a scaling factor based on the Durst curve (Durst, 1960; ASCE, 2005), which relates mean wind speed across different averaging intervals. We use an empirical approximation of the least-squares fitted Durst curve:

$$\Gamma = 1.30 + 0.32 \tanh \left[ 0.79 \log \left( \frac{38}{\Delta_t} \right) \right] \tag{7}$$

where  $\Delta_t$  is the averaging time in minutes. The parameters of these equations are close to those provided in Resio et al. (2002). Using this equation, a scaling factor  $\Gamma = 0.96$  is applied to convert 5-minute mean wind speed to an equivalent 10-minute average.

After standardising the averaging time, we calculate the bias and root mean square error (RMSE) between the NORA3 and lidar data. Outliers are removed by excluding points where the difference between NORA3 and lidar exceeds 10 m s<sup>-1</sup>. The WindCubeV2 and WindCube100S both show a positive bias below 150–200m, indicating that NORA3 tends to overestimate wind speed at lower heights. This overestimation may be due to the influence of the wind farm, which decelerates the flow but is


**Figure 11.** Time series of wind speed at 100 m above the surface during the LOLLEX campaign, comparing the ship—based lidar measurements with the NORA3 reanalysis data. The blue line represents 10-minute averaged wind speeds from the WindCubeV2 lidar wind profiler. The orange line shows 5-minute averaged DBS wind speed data from the WindCube100S scanning lidar. The green line corresponds to hourly wind speed from NORA3. The top panel shows data from 2022, and the bottom panel shows data from 2023.

not represented in the NORA3 model. At higher altitudes, the bias becomes nearly constant at around  $-0.2 \,\mathrm{m \, s^{-1}}$ . The RMSE between the WindCube100S and NORA3 is relatively constant with height, with typical values around  $2 \,\mathrm{m \, s^{-1}}$ . The magnitude of the bias and RMSE is consistent with earlier validation studies of NORA3 in offshore and coastal regions (Cheynet et al., 2025). A local maximum appears near the turbine tip height, possibly due to increased flow variability, not captured by NORA3. The RMSE is slightly higher for the WindCubeV2, which may be caused by uncertainties introduced during motion correction. Unlike the WindCubeV2, the WindCube100S does not undergo full motion correction, except for static tilt correction during vertical stare mode.

Overall, both lidars show consistent performance compared to NORA3, especially above the rotor layer. These results support the use of NORA3 as a reference dataset and provide the necessary foundation for the case studies in the next section.

**Figure 12.** Comparison between the NORA3 reanalysis data and ship—based Doppler wind lidar measurements at various heights during the LOLLEX campaign. Left: Mean bias between NORA3 and the WindCube100S scanning lidar (blue) and WindCubeV2 lidar wind profiler (orange), based on 10-minute averaged wind speed. Right: Root mean square error (RMSE) between NORA3 and the two lidar systems, showing the typical deviation in wind speed at each height.

## 5 First results

This section presents four selected case studies, each highlighting the potential and limitations of the chosen measurement setup.

The first three case studies rely on the dual-lidars deployment on the CTV, focusing on a detailed characterisation of the marine atmospheric boundary layer and its potential interaction with a wind farm, while the fourth focuses on the lidar deployed on the transformer platform and its potential for wake detection and characterisation.

## 5.1 Case study 1 – Kelvin–Helmholtz billows observations

This case study was presented in detail in Malekmohammadi et al. (2025), but the analysis was limited to the period between 12:35 and 13:00. Here, we expand upon that work to highlight the influence of CTV motion on velocity retrieval and to provide recommendations for the use of a scanning DWL on a moving vessel. For completeness, we summarise below the context in which Kelvin–Helmholtz billows (KHBs) were observed, but refer the reader to Malekmohammadi et al. (2025) for further details.

KHBs are wave-like structures caused by shear instability between air layers moving at different speeds (von Helmholtz, 1868; Stull, 1988). They typically form under hydrostatically stable conditions when vertical wind shear exceeds a critical threshold. This leads to dynamic instability and enhanced turbulent mixing, particularly vertical momentum transport. During the LOLLEX campaign, the scanning DWL on the CTV observed KHBs on several occasions. One of the clearest events



**Figure 13.** Time series of the Carrier-to-Noise ratio (CNR) and vertical velocity component *w* recorded by the WindCube100S in vertical stare mode from 12:35 to 13:30 on 2023-02-22. The bottom panel shows the CTV absolute speed, roll and pitch angles during the observational period. The red box indicates the period during which the KHBs were observed.

occurred on 2023-02-22, between 12:30 and 13:00. The KHBs were captured in both the CNR and radial velocity data, as shown in Fig. 13. The time series reveals distinct KHB structures between 12:46 and 12:56 at altitudes ranging from 550 to 750 m. These structures evolved from small wave—like perturbations into billows that grew, overturned, and eventually dissipated. In the vertical velocity field, they appear as alternating zones of upward and downward motion. Before the event, a bright and persistent CNR maximum around 600 m indicates aerosol accumulation beneath the capping inversion, a common precursor to shear—driven instabilities. The KHBs were advected across the lidar beam over approximately 11 minutes, and their passage caused vertical mixing. This mixing is visualised by the broadening and dimming of the CNR maximum, which progressively descended downward to around 400 m by 13:00. However, a local maximum in the CNR remained visible until 13:30, suggesting that elevated aerosol concentrations persisted at around 400 m altitude, even after the dissipation of the KHBs.

Coincidentally, the lidar instruments also captured the near wake of a wind turbine located approximately 37 m upstream. The wake appeared in the velocity time series as a band of enhanced positive vertical velocity between 100 m and 200 m, visible between 12:35 and 13:12, indicating a flow distortion effect by the turbine, potentially from the signature of blade tip vortices. This interpretation is consistent with previous LES-based studies of near-wake dynamics (e.g., Lanzilao and Meyers, 2024). SCADA data confirm that the turbine was operational during the observation period, with a nacelle orientation of approximately 167°, placing the lidar nearly directly downstream of the rotor.





It is important to remember that DWL measurement uncertainties generally increase in the near—wake of turbines, as the Doppler spectrum broadens, complicating the accurate wind speed retrieval (Cheynet et al., 2017a). For the determination of wind profiles from subsequent measurements along different line—of—sight beams, the algorithms also rely on the assumption of homogeneous flow conditions, which are violated in regions of strong flow heterogeneity, such as the near wake of a wind turbine. As such, the observed variability in the lower 200 m of the profile likely reflects increased uncertainty, and results within this region should be interpreted with caution, particularly in the context of turbulence analysis.

The bottom panel of Fig. 13 shows the vessel speed, roll, and pitch angles. Between 12:35 and 13:10, the CTV remained stationary with minimal roll and pitch. After 13:12, the vessel began moving away from the turbine, which led to a negative velocity bias in the WindCube100S data and the disappearance of the near-wake signature. The vessel speed increased to approximately  $4 \text{ m s}^{-1}$  before becoming stationary again near 13:20, after which it resumed motion to a new location.

When the CTV is in motion, the along-beam velocity data retrieved by the WindCube100S becomes significantly noisier, likely due to the combination of high translational speed and varying pitch and roll angles. This limits the instrument's ability to capture accurate velocity measurements and reduces the amount of usable data during periods of vessel motion. Consequently, operating the CTV in stationary mode is preferable when using scanning lidar for turbulence analysis, as it improves both the quality and availability of the retrieved data. Interestingly, the CNR data appear to be less affected by vessel motion. For instance, the dissipation of the KHBs into smaller wave—like patterns after 13:00 remains detectable beyond 13:10, even though the vessel was moving and the data were considerably noisier. This highlights the utility of the CNR as a complementary data source, particularly under non-ideal motion conditions, for tracking coherent structures like KHBs.

The observation of this KHB event revealed a significant increase in the variance of the vertical velocity component within a 400-m thick atmospheric slab, well above the region influenced by the turbine near-wake. In this case, no direct interaction between the KHBs and the wake was observed. However, such interactions may occur under different conditions, e.g. in a shallower ABL, within large wind farms where the internal boundary layer reaches the ABL top, or when KHBs form at lower altitudes, as briefly documented by Radünz et al. (2025). As highlighted by Malekmohammadi et al. (2025), the enhanced vertical mixing associated with KHBs, indicated by elevated vertical velocity variance, could accelerate wake recovery by entraining higher-momentum air downward. We therefore hypothesise that if KHBs overlap more directly with wind turbine wakes, they may enhance wake recovery under stable stratification. This hypothesis warrants further investigation.

# 5.2 Case study 2– Entrainment observations

The second case study documents vertical momentum entrainment during the same day (2023-02-22), but through a different mechanism, i.e., significant downward turbulence motion from the top of the ABL at ca. 1000 m height to the lowest 200 m above the surface. Such measurements were collected before the vessel entered the wind farm, under south-east wind conditions. The CNR profiles indicate a local CNR maximum between 950 m and 1000 m, suggesting an ABL depth of approximately 1000 m. The WindCubeV2 and WindCube100S lidar DBS data revealed a local wind speed maximum of 5.9 m s<sup>-1</sup> around 200 m above the surface. Similarly, the NORA3 model captured a local maximum of around 5.5 m s<sup>-1</sup> for wind speed at approximately 150 m. A significant negative gradient in wind speed was observed above this local maximum. The lidar data



Figure 14. Carrier–to–Noise Ratio (CNR, top panel) and measured vertical velocity component w (middle panel) recorded by the Wind-Cube 100S on 22 February 2023 between 06:35 and 07:16 in the harbour. The bottom panel shows the estimated moving standard deviation,  $\sigma_w$ , of the vertical velocity component w during the same period.

in the top two panels of Fig. 14 reveal large vertical coherence structures characterised by alternating positive (upward) and negative (downward) vertical motions, indicative of significant vertical movements within the ABL.

The profile of the vertical standard deviation of the velocity component w, denoted as  $\sigma_w$ , is shown in the bottom panel of Fig. 14. Between 100 m and 800 m,  $\sigma_w$  exhibits fairly uniform values until 06:46 with a vertical turbulence intensity  $I_w$  under 2% at heights between 100 and 700 m above the surface. Starting at 06:47, a local increase in  $\sigma_w$  becomes visible at an altitude of approximately 600 m. Over time,  $\sigma_w$  continues increases steadily, reaching a peak of approximately 0.5 m s<sup>-1</sup> between 400 m and 600 m above the surface. Between 06:47 and 07:00,  $I_w \approx 0.05$  at 500 m and between 07:05 and 07:12, the turbulence intensity has nearly doubled with  $I_w \approx 0.09$  at 500 m. It should be noted that the lidar beam spatial averaging leads to an underestimation of the turbulence intensity  $I_w$  by 20-30%. Thus, the turbulence intensity increase within one hour is significant. Figure 14 illustrates that this phenomenon reflects a non-stationary and non-homogeneous ABL with a negative vertical motion starting from the top of the ABL near 06:47.

The instantaneous profiles of the vertical velocity and the CNR further suggest enhanced mixing due to turbulence. Measurements were conducted in the morning under stable atmospheric conditions, as indicated by the Profiles of Richardson Number  $(Ri_B)$  (see Malekmohammadi et al. (2025)). Despite the overall stability, the observed increase in turbulence intensity suggests that external factors may have triggered localised mixing events near the top of the ABL around 06:47. A plausible explanation for the enhanced turbulence observed near the ABL top is entrainment of momentum from the free atmosphere, potentially






driven by shear instabilities or interactions with stratocumulus clouds. A strong negative wind speed gradient, recorded by the DWLs and the NORA3 model, supports the presence of shear–driven mixing.

Radiative cooling at the top of stratocumulus clouds can trigger buoyancy–driven turbulence that propagates downward, contributing to enhanced entrainment near the top of the cloud-topped boundary layer (Wood, 2012). This region is typically characterised by strong gradients in humidity and temperature, which are known to support mixing processes. Following Hogan et al. (2009), turbulence dominated by cloud–top radiative cooling is typically associated with negative skewness above the surface layer. In the present case, a negative skewness of vertical velocity, reaching values as low as -0.8 at 300 m height, is observed between 07:05 and 07:15. In contrast, this negative peak is not present in the data from 06:35 to 06:45, before the local increase in the standard deviation  $\sigma_w$ . Between 06:47 and 07:00, the skewness profile is relatively noisy but shows slightly negative values (down to -0.4) between 400 and 700 m. These observations are consistent with the findings of Hogan et al. (2009), supporting the interpretation that the turbulence is amplified from the top of the ABL due to radiative cooling. Although we did not have access to humidity or temperature profiles during this event, the vertical velocity and mean wind speed profiles indicated the presence of strong shear. The exact mechanisms driving the observed entrainment and mixing remain uncertain and warrant further investigation.

These observations document how turbulent entrainment processes can develop under stable morning conditions and how vertical momentum can be redistributed over several hundred meters, even in the absence of convective forcing or classical coherent structures such as KHBs. The non–stationary nature of the turbulence and its vertical extent suggest that entrainment at the ABL top was likely driven by shear instabilities and potentially cloud–top cooling. These findings further underscore the value of using a scanning Doppler wind lidar, complemented by a profiler DWL, on vessels to study entrainment under stable atmospheric conditions. Joint measurements of humidity and temperature profiles using a co-located passive microwave radiometer would be valuable for distinguishing between shear– and buoyancy–driven entrainment mechanisms. Additionally, ultrasonic anemometers could usefully complement the lidar data by providing near–surface turbulence measurements.

#### 410 5.3 Case study 3 – Internal wave observations

The evening of 22 February was characterised by stable stratification, a wind speed of 10 m s<sup>-1</sup> at 100 m, a strong near–linear shear of approximately 0.06 m s<sup>-1</sup> m<sup>-1</sup> between 40 m and 100 m and a cloud cover above 600 m as indicated by the CNR profiles. The WindCube 100S detected the presence of a low–level jet, with a maximum wind speed of 15 m s<sup>-1</sup> between 420 m and 450 m above the surface. From 22:05 to 22:30, the lidar system recorded wind conditions suggestive of internal atmospheric waves below the cloud-topped ABL. Such waves are often associated with the presence of low–level jets (Jia et al., 2019). In the present case, the observed wave–like motions may have been triggered by shear instability near the maximum of the low–level jet, where vertical wind shear was strongest. Wave patterns began to form at 22:16 and became visible in both the CNR (above 400 m) and the vertical velocity fluctuations (from 200 m to 600 m). It should be noted that these waves were not generated by wind farms and are therefore distinct from the so–called farm–generated gravity waves (Allaerts and Meyers, 2018). When internal atmospheric waves, including gravity waves, form upstream of a wind farm and at a height low enough to interact with

**Figure 15.** Vertical profile of the observed wave–like pattern on 22 February 2023. Top panel: Time series of the carrier–to–noise ratio; middle panel: instantaneous vertical velocity fluctuations; bottom panel: instantaneous standard deviation along the scanning beam. The wave–like pattern is visible between 200 m and 600 m altitude from 22:16 to 22:30.

the farm's internal boundary layer, they may enhance wake recovery by amplifying mixing and wake meandering (Feng and Watson, 2025).

Figure 15 shows the vertical structure of these oscillations, which span approximately 200 m in height. The wave–like motions are visible both in the instantaneous vertical velocity (middle panel) and the corresponding standard deviation (bottom panel). The internal wave structure coincides with the nose of the low–level jet. A spectral analysis of the vertical velocity at 200, 300, 400, 500, and 600 m revealed a dominant periodic signal with a 2-minute cycle, as shown in Fig. 16. The observed phase shift between altitudes suggests that vertical wind shear strongly influences the wave structure. The spectral peak falls within the mesoscale range, rather than the typical turbulence range, and is consistent with internal atmospheric waves rather than turbulent eddies.

The observed period places these waves at the lower end of the internal gravity wave spectrum. For comparison, Banakh and Smalikho (2016) reported oscillation periods between 6.5 and 18 minutes in coastal terrain, while Jia et al. (2019) observed shear—driven gravity waves with periods exceeding 10 minutes. According to Sun et al. (2015), internal gravity wave periods typically range from 1 to 30 minutes, indicating that the 2-minute period observed here is short but still within the expected range.

As shown in the bottom panel of Fig. 15, the local increase in the standard deviation of vertical velocity extends to heights below 200 m, even though the CNR indicates that the wave–like pattern is strongest around 500 m. Given the proximity of the

445

**Figure 16.** Power spectral densities (top panel) and associated time series (bottom panel) of the vertical velocity fluctuations from 200 to 600 m above the surface on 22-02-2023 from 22:16 to 22:30.

measurement site to the coast and its distance of about 20 km from the nearest offshore wind farm, it is possible that similar wave—like motions could occur above the wind farm itself. If so, such waves could enhance wake recovery by amplifying vertical momentum entrainment into the farm, especially considering that the internal boundary layer generated by the wind farm is likely confined to below 200 m.

The previous case studies, along with this one, demonstrate the ability of scanning and profiling DWLs to capture dynamic processes in the ABL, including momentum entrainment associated with internal waves. However, limitations remain for a complete characterisation of internal atmospheric waves. In particular, the absence of co-located temperature and humidity measurements restricts our ability to study the stratification of the atmosphere in detail and, therefore, to identify the dominant wave generation mechanisms. While wind speed profiles confirm the presence of a low–level jet and support the interpretation of shear-induced wave activity, additional observations would be needed to distinguish between shear- and buoyancy–driven processes.

# 5.4 Case study 4 – Wake observations

This case study presents preliminary results from RHI scans performed by the long-range scanning DWL installed on the transformer platform north of Rødsand II. Two representative cases are shown: one with easterly winds (Fig. 18) and one with

485

westerly winds (Fig.19). Hereinafter, we focus on vertical slices of the mean along-beam velocity component in the wind turbine wake. For comparison, virtual RHI scans were generated using PyWake to assess the reliability of the measurements. The PyWake simulations were initialised using mean wind speed and direction from NORA3, assuming undisturbed inflow conditions. For the easterly wind case, wind directions from NORA3 and the lidar profiler (WindCubeV2) on the CTV were in close agreement. However, for the westerly wind case, the lidar profiler data were not reliable enough to validate the NORA3 direction.

In PyWake, the wind farm layout was modeled using the PropagateDownwind model combined with the TurboGaussianDeficit wake deficit model, which is similar to Ørsted's TurbOPark model. Our choice of this model is motivated by Fischereit et al. (2022), who found that analytical wake models (AWMs) tend to underestimate the wake deficit, i.e. they overestimate wake recovery between the Rødsand II and Nysted wind farms. The TurboGaussianDeficit model, in contrast, typically produces significantly longer wakes than other Gaussian wake deficit models such as those by Zong and Porté-Agel (2020) or Niayifar and Porté-Agel (2016), and we consider it more suitable for the present case. We note that our results differ from those reported by Souaiby and Porté-Agel (2024), who did not use PyWake. These differences may reflect variations in how the analytical wake models were implemented and configured.

The simulations also included the GaussianOverlapModel for rotor-averaged wind speed, the mirror ground model, the Crespo-Hernandez turbulence model, and a hub-height turbulence intensity of 0.1. Finally, wake superposition was modeled using the squared-sum method.

Figure 18 compares lidar and PyWake results for the case on 2023-05-20, between 05:13 and 05:28 UTC, with an azimuth of 206°. NORA3 and lidar profiler reported mean wind speeds of 9.0 m s<sup>-1</sup> and 7.8 m s<sup>-1</sup>, respectively, both with a direction of 76° at 100 m above the surface. Satellite data (Fig.17) confirm a north-easterly flow, with surface wind speeds exceeding 8 m s<sup>-1</sup> upstream of the wind farm, indicating that the actual upstream wind speed may have been higher than measured by the lidar profiler, which was located inside the farm.

Figure 19 presents a similar comparison, but for a westerly flow with a wind speed of  $9.0 \text{ m s}^{-1}$  at hub height and a wind direction of  $278^{\circ}$  on 2023-06-01. In both cases, the direction was adjusted from the NORA3 wind direction by a value of  $-6^{\circ}$  to better align the position of the first wake centreline with the lidar observation and to provide a comparable estimate of the along-beam velocity component.

For easterly winds, the lidars can capture both wakes from individual turbines from the Rødsand II wind farm but also the farm-induced wake from the Nysted wind farm, which is known to lead to additional wake loss in Rødsand II (Hansen et al., 2015; Van der Laan et al., 2015). For westerly winds, no farm clustering effect takes place. In both cases, the RHI scans from the lidar and the simulations reveal multiple turbine wakes. For the easterly flow, the wake of the first turbine, located about 0.8 km from the lidar, is captured. However, the alignment of the second and third wakes (1.0 km and 1.5 km away) between measurements and simulations is less accurate. This mismatch may result from minor variations in wind direction, which can significantly shift the wake centre in virtual scans. Additional sources of error may include wind veering (Bodini et al., 2017), yaw misalignment due to turbine control strategies and maybe the Coriolis force effect on the wake of the Nysted wind farm (Van der Laan et al., 2015).

**Figure 17.** Surface wind speed from SAR on 2023-05-20 at 05:33:25. Red dots indicate turbine locations in the wind farms near Lolland, taken from the database by Zhang et al. (2021). The image shows north-easterly flow with surface wind speeds exceeding 8 m s<sup>-1</sup> upstream of Rødsand II.

In the westerly flow case, two wakes are observed at approximately 500 m and 1.2 km. The wake at 1.2 km appears broader and may result from the superposition of multiple wakes, as indicated by the PyWake simulation. Between these two wakes, PyWake predicts additional far wakes that are not clearly visible in the measurements. As with the Easterly case, small changes in wind direction have a noticeable effect on the simulated wake position and velocity field.

Differences in wind speed between simulations and measurements likely originate from both wind direction uncertainty and underestimated upstream wind speeds. During the campaign, upstream conditions were not systematically measured, as the vessel-mounted lidars were located in a sheltered harbour environment or inside the farm most of the time. Overall, several thousand RHI scans were collected during the campaign. We believe this dataset is valuable for further validating AWMs in the Rødsand II wind farm. Although a qualitative analysis suggests that the TurbOPark model performs well in our case, further work is needed to refine inflow conditions and identify possible yaw misalignment using SCADA data. In particular, future studies should include a systematic quantitative error analysis, using e.g. the root-mean-square error or bias between the observed and virtual RHI scans. Such a comparison would help identify the most suitable wake deficit models. Also, such RHI scans may complement nacelle-mounted lidars for wake analysis, which could be the topic of further study.

# 6 Discussion

490

500 CTV-based measurements are subject to operational limits. Measurements are only possible during daytime working hours, typically from 07:00 to 19:00. This limits the ability to capture stable nighttime conditions and diurnal transitions. In addition,

515

Figure 18. Comparison of 14-min mean along-beam wind speed on 2023-05-20 for an azimuth of 206°. Top panel: Measured by the scanning lidar (05:14-05:28). Bottom panel: Virtual RHI scan from PyWake, using inflow conditions of 9.0 m s<sup>-1</sup> wind speed and 70° direction at 100 m altitude.

safety regulations prohibit CTV operations when significant wave heights exceed 2 m or wind speeds surpass 12 m s<sup>-1</sup>. These thresholds reduce data availability during strong-wind or swell events. In contrast, buoy-based lidars can operate continuously, including during high-wind conditions, though they lack the flexibility to capture different spatial locations.

The WindCubeV2 lidar showed robust performance on the moving vessel. Its design for offshore conditions, including reduced mechanical complexity and compatibility with motion correction, contributed to stable operation. The WindCube 100S scanning lidar, in contrast, was more sensitive to translational motion. This sensitivity caused interruptions in some scans or led to incomplete datasets. Despite this, the system still produced many high-quality scans suitable for wake and turbulence analysis.

Power supply considerations are an important factor when installing lidar systems on offshore vessels. Each vessel has a 510 different electrical configuration, which influences how equipment can be safely integrated. In this campaign, the CTV required a daily power reset, which posed a risk to the WindCube100S due to its higher power demands and limited internal buffering. To address this, the lidar was connected to an external uninterruptible power supply (UPS) housed in a watertight enclosure. This configuration ensured stable operation during brief power interruptions. Future deployments can benefit from standardised power interface protocols and pre-tested UPS systems to improve installation efficiency and system robustness across different vessel types.

525

530

**Figure 19.** Comparison of 14-min mean along-beam wind speed on 2023-06-01 for an azimuth of 206°. Top panel: Measured by the scanning lidar (16:28–16:42). Bottom panel: Virtual RHI scan from PyWake, using inflow conditions of 9.0 m s<sup>-1</sup> wind speed and 278° direction at 100 m altitude.

In principle, a dual-lidar setup like the one tested here could also be installed on a buoy. However, this would require careful consideration of platform stability and instrument design. Scanning DWLs are physically larger and more mechanically complex than profiler lidars, and most commercial models do not include active motion correction. Operation on a floating platform would therefore necessitate additional stabilisation measures or dedicated post–processing procedures. While motion correction is routinely applied to profiler lidars on buoys, extending this capability to scanning systems requires further development. Alternatively, vessels such as CTVs provide a more stable and spacious platform. Among available offshore platforms, the CTV offers a cost-effective and mobile solution for short-term campaigns in shallow coastal areas.

Finally, the success of this campaign was supported by favourable environmental conditions. The measurement site was located in a shallow-water region with low average wave heights. The surrounding coastal topography and the short distance to the harbour enabled nearly daily access to the wind farm. These factors made it feasible to implement a mobile lidar strategy using a CTV. In deeper waters or more remote offshore locations, this approach would require significant adjustments. Vessels may not be able to visit the site daily, resulting in fewer data collection opportunities. Harsher sea states would increase platform motion, leading to greater uncertainty in the retrieved wind velocities due to tilt and translation effects. The exposure to rough conditions may also increase the risk of contamination or damage to the lidar optics, for example, due to sea spray on the scanner head.

27

To address these challenges, future campaigns may consider integrating real–time motion correction systems. One promising approach could involve gyroscopic self-levelling platforms as discussed in Malekmohammadi et al. (2024). Such systems could minimise the impact of platform tilt and reduce the reliance on post–processing corrections. While these systems may not be practical for small buoys due to size and power limitations, they could be feasible on larger vessels or semi-permanent floating platforms. Further investigation into active stabilisation methods for scanning lidars could help extend mobile measurement strategies to more demanding offshore environments.

#### 7 Conclusions






The study presents new measurements of vertical momentum entrainment above an offshore wind farm, collected during the LOLland offshore Lidar EXperiment (LOLLEX). The campaign introduced a novel dual-lidar setup mounted on a crew transfer vessel (CTV), combining a scanning and a profiling Doppler wind lidar. This mobile platform enabled flexible sampling of the atmospheric boundary layer (ABL) at various positions and under different wind directions. To complement these mobile observations, a long-range scanning lidar was installed on a transformer platform at the northern edge of the Rødsand II wind farm. This fixed lidar provided continuous, high-resolution range—height indicator (RHI) scans along a fixed line of sight across the wind farm. Together, the two systems allowed for the study of entrainment processes and wake recovery up to 3 km above the surface. The campaign lasted one year and produced several thousand hours of Doppler lidar data.

To illustrate the potential and limitations of this measurement setup, we presented four case studies. These examples highlight different entrainment-related processes and demonstrate the value of combining scanning and profiling Doppler wind lidars on both mobile and fixed platforms:

- Kelvin-Helmholtz billows were captured at altitudes between 550 and 750 m, indicating increased vertical mixing under stable conditions. This case illustrates how the vertical stare mode can be used to quantify turbulence statistics associated with shear instabilities (Malekmohammadi et al., 2025), but also reveals how vessel motion can significantly degrade the quality of scanning lidar data.
- 2. A case of downward turbulent mixing was observed during one morning, where momentum was transferred from the top of the ABL to lower levels in the absence of classical structures like KHBs. This transient phenomenon, which may involve cloud-top cooling, shows that vertical entrainment can occur deeper into the boundary layer than documented in the first case study.
- 3. Internal atmospheric waves with a 2-minute period were detected near the nose of a low–level jet. These shear-induced oscillations suggest that entrainment processes may not be limited to turbulence but also include wave-driven mechanisms, particularly under stable atmospheric conditions.
- 4. Wake profiles inside the wind farm were recorded using RHI scans from the platform-mounted lidar. These were compared with PyWake simulations, offering valuable data for wake model validation. This case demonstrates that the campaign



was not limited to turbulence studies but also supports direct observations of wake deficits, which are an essential part of understanding wake recovery.

Together, these results show that scanning lidars and lidar wind profilers can detect dynamic ABL processes relevant to wake recovery. They also demonstrate the value of mobile lidar systems for offshore measurements, especially when flexibility in spatial coverage is needed. Still, several limitations remain. The scanning lidar was highly sensitive to vessel motion, and data quality degraded significantly while the CTV was in transit. Motion correction was only applied in post–processing to the profiler lidar. No real–time correction was implemented, real–time stabilisation or onboard correction would likely improve performance, particularly if deployed on a larger, more stable platform. The use of a CTV also restricts operations to relatively calm sea states (significant wave height  $H_s < 2$  m) and moderate wind speeds ( $\overline{u} < 12$  m s<sup>-1</sup>), limiting data availability. In addition, the lack of co-located temperature and humidity measurements prevented a full characterisation of atmospheric stratification, making it difficult to separate buoyancy– and shear–driven processes.

The dataset collected during the LOLLEX campaign represents a valuable resource for wind energy science and boundary layer meteorology. Future studies using this dataset could explore the following:

- 1. Adding instruments such as microwave radiometers, radiosondes, or ultrasonic anemometers could help describe the vertical temperature structure and near-surface turbulence more accurately. This is important for understanding atmospheric stability and mixing over the sea.
  - 2. Using real-time motion correction or stabilised platforms could increase the amount of usable lidar data and improve the quality of turbulence measurements from moving platforms. While stabilised platforms are challenging to implement on buoys due to size and power constraints, they may be more feasible on vessels thanks to greater space and onboard power supply.
  - 3. A more detailed evaluation of near-wake measurements is needed. Case study 1 showed that vessel-mounted lidars can detect near-wake structures. The LOLLEX campaign provides a unique opportunity to investigate this further, as it includes numerous high-resolution scans collected at varying distances and angles relative to turbine wakes.
- 4. Combining data from the fixed platform lidar and the vessel-mounted lidars. This could give a more complete picture to understand wind turbine wake recovery by using wind velocity data across the rotor plane, both in the horizontal and vertical directions.
  - 5. Use vertical stare scans from the scanning lidar to estimate the atmospheric surface layer depth. This approach can follow methods such as those proposed by Puccioni et al. (2024). Accurate estimation of ABL depth is important for modeling wake recovery and vertical momentum entrainment in offshore wind farms. Comparing these lidar-based estimates with boundary layer depths from reanalysis datasets would help assess the reliability of reanalysis products in coastal regions of Northern Europe.



Data availability. WindCubeV2 lidar wind profiler metadata are available in the Figshare repository: https://doi.org/10.11583/DTU.22739729. v1. The full datasets from both the WindCubeV2 and the WindCube100S scanning lidar are currently being processed. Due to the large volume of data, only the subsets used in the case studies presented in this paper will be made publicly available in the near future. The complete datasets are available from the authors upon request.

Author contributions. Conceptualisation of the campaign was done by JR, GG, CL, JM, MS, and SM. Data analysis and creation of figures was done by SM and EC. Model simulations with PyWake were performed and analysed by EC. Project management and funding was handled by GG and JR. The lidar deployment on the vessel was prepared and supervised by SM. The original draft was prepared by SM, EC and JR, with contributions by JM and MS and the review and editing were done by EC, JR, GG, CL, JM and MS.

Competing interests. The authors declare no conflict of interests.

Acknowledgements. The LOLLEX campaign was funded by the European Union Horizon 2020 research and innovation program under grant agreement no. 861291 as part of the Train<sup>2</sup>Wind Marie Sklodowska-Curie Innovation Training Network (https://www.train2wind.eu/). The OBLO (Offshore Boundary Layer Observatory) project, funded by the Research Council of Norway (project number: 227777), provided the lidar instrumentation. We gratefully acknowledge the owners of Rødsand II for the access to the wind farm and the support of RWE for providing access to one of the crew transfer vessels from Northern Offshore Service, enabling the success of our campaign. Special thanks to the involved staff at RWE Wind Services Denmark and RWE Offshore Wind, Innovation & Industrialization. The authors would also like to express their gratitude to Christiane Anabell Duscha, Anak Bahadur Bhandari, Tor Olav Kristensen at UiB, and Per Hansen, Gunhild Rolighed Thorsen, Kasper Clemmensen, and Elliot Simon at DTU for their unconditional technical and logistical assistance.

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
