# Peer review of "The LOLland offshore Lidar EXperiment (LOLLEX): A novel observational approach for the study of wind farm flow and entrainment"

_EGUsphere, 2025_

## Referee Comment (RC1)

**Comments to EGUSPHERE-2025-3148**

**General comments**

The paper presents an interesting dataset collected around an offshore wind farm with a combination of three lidars. The results are mostly qualitative and not novel (i.e., atmospheric waves occur over the ocean) and present some limitations. The authors create high expectations in the abstract and introduction by referring to a novel "strategy to detect vertical momentum entrainment", but it falls a bit short in the content.

First, for momentum entrainment one usually refers to the $\overline{u'w'}$ Reynolds stress, while only $w'$ is shown. Then, the only novelty in the strategy seems to be the fact that the lidars are on a ship yet performing well-known scans. The potential of the colocation of a V2 and a 100S is not explored as only measurements from individual instruments are shown.

This brings to the second point: none of the case studies show profiles of wind speed and direction, which would be interesting to see after all the efforts in motion correction and comparison with NORA3.

Third, it sounded like the wind profiles from the 100s were not motion-corrected. This is hard to understand and the fact that the RMSE of the 100S winds have lower RMSE than the V2 ones when compared to NORA3 creates additional confusion in the reader.

Fourth: another singular approach is the execution of RHIs trying to capture the wake velocity in the cross-stream plane. Although the lidar is positioned favorably to scan the y-z plane for E-W wind, the line-of-sight projection of the wake velocity field is minimal. This counters almost all the lidar wake campaign (ground- or nacelle-based) where indeed one tries to align the lidar with the wind as much as possible.

Acknowledging these limitations, major revisions are recommended. A revised version of the paper should address these four points:

- Either estimate $\overline{u'w'}$ with a new approach or mitigate the claims of a novel strategy.
- Show wind speed and direction from both lidars for the case studies.
- Motion-correct the 100S profiles or justify why this is not done.
- Explain better the quite counter-intuitive RHI for cross-stream wake detection strategy or remove that section (it adds little information on the wake physics and model accuracy).

**Specific comments**

L11: which scanning lidar was doing profiling + stares? Please specify.

L36: I would replace "beyond certain scales" with "for large wind farms".

L53: What does "most offshore DWLs rely on fixed platform"? For installation or for validation? Please rephrase.

L101: Please specify that the WindCube are the lidars on the ship or just avoid mentioning the lidar model at this point.

L129: IMU-derived location can drift over time, was it corrected with the GPS? Please expand.

L133: 5 minutes is shorter than the typical 10 minutes used by the wind industry, please justify this choice.

Fig.4: Please keep same font size across the figure.

Section 3.1: "filtering" and "outlier removal" are generally synonyms. Please clarify if this is simple a 2-step quality control using first SNR and then median filter.

Section 3.2: Please provide information on the timestep used in the correction.

Eq 6: This equation uses only the tilt angle while for sonic anemometers and in reality also pitch and yaw play a role in defining the projection of the horizontal wind speed onto the beam direction. It other words, this seem to apply to the special case where the tilt is plane defined by the wind direction. Please justify this simplification.

L243: Vertical wind speed is never "removed" in turbulence analysis, please omit.

Fig. 11: Please make time format consistent between the two panels.

L298: Please mention that NORA may not be capturing the physics of the near-surface winds (low-level jet, sea surface temperature, wave height, etc.), on top of the wake effects. It is not uncommon for NWP to show biases in the lower ABL.

L315: Please add the date and the time zone, not just the hour.

L334: It is unclear why tip vortices would have a persistent positive velocity instead of a +/- pattern. Also, it is questionable that the resolution of the lidar would allow to resolve a tip vortex. Please either omit or add more references showing the same positive w over such a large vertical range.

L339: The statement about the need for horizontal homogeneity to derive wind profiles is correct, but horizontal wind is not shown in Fig. 13 (only w, which does not require this assumption). Please clarify.

L367: Please clarify if the vessel was moving during the case study 2.

Section 5.2: The KHB hypothesis seems to be ruled out too quickly without a clear justification. In fact, at L388 the "shear instability" is mentioned. Please reconcile.

L505: it is unclear where the robustness of the motion correction of the V2 as shown. The V2 showed instead higher RSME when compared to NORA3. Please clarify or omit.

Discussion: this section could be combined with the Conclusion since it repeats many concepts and makes the paper unnecessarily lengthy.